# The Tryptophan and Kynurenine Pathway Involved in the Development of Immune-Related Diseases

**DOI:** 10.3390/ijms24065742

**Published:** 2023-03-17

**Authors:** Ai Tsuji, Yuka Ikeda, Sayuri Yoshikawa, Kurumi Taniguchi, Haruka Sawamura, Sae Morikawa, Moeka Nakashima, Tomoko Asai, Satoru Matsuda

**Affiliations:** Department of Food Science and Nutrition, Nara Women’s University, Kita-Uoya Nishimachi, Nara 630-8506, Japan

**Keywords:** tryptophan, kynurenine, immune-related disease, engram, gut–brain axis, gut microbiota, inflammation, reactive oxygen species

## Abstract

The tryptophan and kynurenine pathway is well-known to play an important role in nervous, endocrine, and immune systems, as well as in the development of inflammatory diseases. It has been documented that some kynurenine metabolites are considered to have anti-oxidative, anti-inflammatory, and/or neuroprotective properties. Importantly, many of these kynurenine metabolites may possess immune-regulatory properties that could alleviate the inflammation response. The abnormal activation of the tryptophan and kynurenine pathway might be involved in the pathophysiological process of various immune-related diseases, such as inflammatory bowel disease, cardiovascular disease, osteoporosis, and/or polycystic ovary syndrome. Interestingly, kynurenine metabolites may be involved in the brain memory system and/or intricate immunity via the modulation of glial function. In the further deliberation of this concept with engram, the roles of gut microbiota could lead to the development of remarkable treatments for the prevention of and/or the therapeutics for various intractable immune-related diseases.

## 1. Introduction

Among the 20 amino acids that supply protein formation, tryptophan is one of the essential mammalian amino acids, and the kynurenine pathway is the major metabolic pathway of the tryptophan metabolism, which is known to play an important role in the nervous, endocrine, and immune systems [1]. The tryptophan metabolism is also closely related to inflammation and inflammatory diseases such as infection, coronary heart disease, autoimmune syndrome, cancer, and neurodegenerative disorders [2]. Similarly, immune activation is intricately linked with the dysregulation of the tryptophan metabolism on the way to oxidative breakdown along the kynurenine axis [2]. In addition, the amino acids in the tryptophan–kynurenine pathway have been shown to be associated with the risks of atherosclerotic cardiovascular events, which may predict acute coronary events in older individuals without previous coronary heart disease. [3]. Furthermore, accumulating evidence has also suggested a role of the kynurenine pathway in various diseases and disorders, including Alzheimer’s Disease, depression, schizophrenia, amyotrophic lateral sclerosis, Huntington’s Disease, and/or cancers [4]. The tryptophan–kynurenine pathway is the most important course for the tryptophan catabolism, accounting for around 95% of dietary tryptophan degradation [5]. As a consequence of the kynurenine pathway’s metabolic integration, inflammation could contribute to the accumulation of kynurenine in the host brain, which may be associated with depression and/or schizophrenia [5].

Dependent on the nutritional availability of the essential amino acid tryptophan, bacteria of the gut microbiota could influence the host’s serotonin biosynthesis in enterochromaffin cells. Therefore, certain members of the gut microbiota might be capable of delivering serotonin to their host [6]. An increase in serotonin levels, induced by spore-forming bacteria in the gut, might contribute to the progression of inflammation, with an impact on neutrophil function [7]. An additional tryptophan-dependent pathway that is modulated by the gut microbiota is the indole pathway, which also impacts mucosal Th17 cells’ immunity in the gut via the aryl hydrocarbon receptor (AHR) [8,9]. In addition, the tryptophan–kynurenine pathway might underlie the regulation performed by gut microbiota-regulated interferons [10]. Meaningfully, a protective role of dietary tryptophan has been demonstrated in mouse models [11]. This review provides a novel view of the available evidence of tryptophan and downstream metabolites in several immune-related diseases, in the context of underlying inflammation.

## 2. Tryptophan and Kynurenine Metabolic Pathway

The catabolism of tryptophan has two major enzymatic pathways. The first pathway is the serotonin pathway, which is mediated by tryptophan hydroxylase and results in the generation of serotonin or 5-hydroxytryptamin (5-HT), a precursor of melatonin [12]. In the other pathway, tryptophan is catabolized to kynurenine, the neuroprotective kynurenic acid, and the neurotoxic quinolinic acid (QA) [13]. Interestingly, D-tryptophan can also be enzymatically converted to L-tryptophan by D-amino acid oxidase and aminotransferase (Figure 1). Tryptophan is mainly catabolised through the kynurenine pathway [14], which is controlled at its first step and catalyzed by the rate-determining enzymes, tryptophan 2,3-dioxygenase in the liver and indoleamine 2,3-dioxygenase in the extrahepatic tissues [15]. The tryptophan and kynurenine pathway accounts for the majority of ingested tryptophan [12,16]. Kynurenine could be further metabolized into several other downstream products, such as 3-hydroxykynurenine, xanthurenic acid, anthranilic acid, 3-hydroxyanthranilic acid, and picolinic acid, which are collectively referred to as kynurenines. Kynurenine is converted into anthranilic acid by kynureninase, and to 3-hydroxykynurenine by kynurenine monooxygenase. Then, 3-hydroxykynurenine is metabolized into 3-hydroxyanthranilic acid by kynureninase. Lastly, quinolinic acid is generated, and thereafter, so is nicotinamide adenine dinucleotide (NAD), which plays an essential role in energy metabolism. The pathway involving the indoleamine-2,3-dioxygenase has been assumed to play a major role in the development of major depression. In addition, the tryptophan 2,3-dioxygenase activity is inhibited by the glucose intake in liver, probably involving the increased production of the feedback allosteric inhibitor, NADPH [17]. This may explain the associations between the elevated concentrations of kynurenine metabolites with the impaired glucose tolerance observed in individuals that are overweight or have obesity [18]. In patients with obesity, the levels of 3-hydroxykynurenine and/or 3-hydroxyanthranilic acid appear to be positively correlated to an impaired glucose tolerance [18]. Furthermore, associations of kynurenines with cancers have been reported [19]. The tryptophan and kynurenine pathway may play a key role in inflammation, which is supported by the reported associations of kynurenines with chronic diseases and/or cancer [19].

Some kynurenines are considered to have anti-oxidative, anti-inflammatory, and neuroprotective properties, while others have pro-oxidative, pro-inflammatory, and neurotoxic properties, and others have somewhat less well-characterized properties [20]. The tryptophan–kynurenine pathway is recognized to be involved in quality of life (QOL) after cancer [5,21,22], which may play a crucial role in inflammation [23]. In addition, the kynurenine pathway determines the overall neuronal excitability and plasticity by modulating the glutamate receptors and G protein-coupled receptor 35 (GPR35) activity across the central nervous system (CNS), and regulating the general features of immune cell status, surveillance, and tolerance, which often involves the AHR. The ratio of kynurenine-to-tryptophan may be a recognized marker of cellular immune activation [24]. In contrast, high levels of the ratio of kynurenic acid/quinolinic acid, a ratio of two strong antagonists of N-methyl-D-aspartate (NMDA) receptors, have been considered to reflect neuroprotection [25]. Additionally, there has been a major development of kynurenic acid analogues as neuroprotectants for the treatment of neurodegenerative diseases and stroke [25]. During inflammation, the balance between the potentially neuroprotective and neurotoxic kynurenines is disturbed, which is reflected in their respective ratios.

## 3. Connection between Kynurenine Pathway and Immunity

Inflammatory cytokines could activate indoleamine 2,3-dioxygenase. For example, interferon gamma can activate the enzyme of indoleamine 2,3-dioxygenase, thereby shifting the Trp metabolism to Kyn production [26]. Cytokines and kynurenines are closely associated with mediating the communication between the brain and the immune system, which could regulate the neuron and/or glial activity in the central nervous system (CNS), as well as the function of the immune system within a combined network. This concept would broaden the scope for the development of new treatments for disorders that involve immune systems with harmless and/or more active agents [27]. During inflammation, pro-inflammatory cytokines, mainly tumor necrosis factor-α (TNF-α) and interferon-γ (IFN-γ), may increase the catabolism of tryptophan by encouraging the expression of indoleamine 2,3-dioxygenase [28]. Increasing evidence may verify inflammation as key for the entry of tryptophan into the kynurenine pathway, and, in return, this pathway might metabolite as a crucial modulator of inflammatory responses. The kynurenine pathway might also be activated to generate sufficient amounts of NAD+ energy, as activated immune cells may need a great deal of energy to fight against pathogens [29], which could protect the host against the oxidative stresses that are linked to various inflammations. Significantly, a lot of the kynurenine metabolites may have the potential for immune regulation, which could alleviate the inflammation. For example, kynurenic acid could reduce pro-inflammatory cytokines [30]. In addition, kynurenine and 3-hydroxyanthranilic acid could bring in the apoptosis of Th1 cells and/or natural killer cells [29]. Overall, kynurenine metabolites may be important for energy metabolism and/or immune modulation. Accordingly, the activation of the kynurenine pathway might be important for immune tolerance in pregnancy [31], and the immunosuppressive roles of kynurenine metabolites have been comprehensively reviewed [30]. In these ways, the anti-inflammatory and/or immunosuppressive functions of kynurenine metabolites are of particular interest. Furthermore, tryptophan may be involved in stress responses, such as sleep in order to dismiss these stress responses. However, the kynurenine pathway is dependent on the AHR, which could also lead to cancer progression via the down-regulating T-cell phenotype, to some extent [32]. Interestingly, fluoxetine could improve the metabolic disorder of tryptophan in vivo that is induced by chronic stresses, and may inhibit cancer growth and invasiveness, possibly by suppressing the kynurenine pathway [33]. For example, an increase of kynurenine might contribute to the aggressiveness of cancers, as the activation of the entire kynurenine pathway and/or the subsequent production of NAD^+^ have been shown to promote cancer cell dissemination in triple-negative breast cancers [33]. Pathologically, the kynurenine pathway has been associated with a variety of diseases, including Huntington’s Disease [34], cardiovascular diseases [35], and osteoporosis [36]. Evidence for a pathophysiologically significant role of the abnormal metabolism in the kynurenine pathway might be most convincing in the case of Huntington’s Disease [34]. Indoleamine 2,3-dioxygenase, an enzyme that catalyzes the rate-limiting step in the kynurenine pathway of tryptophan degradation, may be also intensely induced by an inflammation in the artery walls [35]. In addition, the kynurenine pathway is also associated with osteoblastogenesis, which may be linked to the pathophysiology of osteoporosis [36]. The activity of osteoblasts is reduced by 3-hydroxykynurenine, a product of kynurenine [36]. However, their precise mechanisms remain to be elucidated.

## 4. Several Immune-Related Diseases Associated with the Tryptophan and Kynurenine Pathway

Again, the atypical activation of the tryptophan and kynurenine pathway might be involved in the pathophysiological progression of several intricate diseases, including cancers, and some metabolic diseases, including diabetes, obesity, and/or cardiovascular diseases [37,38]. In addition, the relationships of the tryptophan and kynurenine pathway have been well recognized in mental and mood disorders, neurodegenerative diseases, and other inflammatory situations. Here, we would like to introduce some of the representative immune-related diseases which might be associated with the tryptophan and kynurenine pathway.

### 4.1. Major Depressive and Bipolar Disorders

The metabolism of tryptophan, which is the precursor of neurotransmitters, has been deliberated to be one of the important biological pathways for depression [39]. Possibly based on the relationship of the tryptophan metabolism with mental disorders, inflammatory diseases are frequently comorbid with major depression. In general, the dysregulation of tryptophan metabolites, such as serotonin, kynurenine, quinolinic acid, and kynerunic acid, might also be interrelated to depressive disorders. In fact, there is robust evidence that the activation of the immune-inflammatory reaction system and/or the compensatory immune-regulatory system might play an important role in the pathophysiology of major depressive and bipolar disorders [40], which are considered by their elevated production of various cytokines, including interleukin (IL)-6, interferon (IFN)-γ, and tumor necrosis factor (TNF)-α [41]. The activation of the oxidative stress pathway may be associated with the main features of emotional disorders and the severity of an illness [42].

### 4.2. Cardiovascular Disease

Cardiovascular diseases, including myocardial infarction and/or stroke, may constitute a leading cause of death, globally. The pathophysiology of cardiovascular diseases is linked to vascular inflammation [43], which may be linked to hypertension. Aryl hydrocarbon receptor (AHR)-signaling is well recognized to contribute to those cardiovascular pathologies by inducing the expression of pro-inflammatory interleukin IL-1β, IL-8, and TNF-α, with the following foam cells [44]. Therefore, the tryptophan metabolites may also have an influence on vascular inflammation [45]. There are comprehensive experimental and/or clinical suggestions for the involvement of microbiota-related tryptophan metabolites in the vascular inflammation, hypertension, and/or development of cardiovascular diseases [46]. In addition, the kynurenine/tryptophan ratio may be positively associated with advanced atherosclerosis in clinical studies [47].

### 4.3. Kynurenine Pathway in Acute Kidney Injury and Chronic Kidney Disease

Kidney diseases, including acute kidney injury and/or chronic kidney disease, may represent a worldwide health issue [48,49,50]. The kynurenine pathway might also be activated in acute kidney injury [51] and the kidney tissues of mice with ischemia/reperfusion-induced acute kidney injury [52]. Consistently, mice with acute kidney injury arising from drug-induced toxicity might have a higher amount of kynurenine [53]. Mechanistically, the upregulation of the kynurenine metabolism during acute kidney injury may be explained by the increase in TNF-α and IFN-γ, which are robust activators of indoleamine 2,3-dioxygenase. Consequently, the upregulation of the kynurenine pathway during acute kidney injury and/or chronic kidney disease might be attributed to the production of inflammatory cytokines. Some enzymes, such as kynurenase in the downstream kynurenine pathway, could result in the further accumulation of intermediates [54], which may also contribute to the pathogenesis of acute kidney injury and/or chronic kidney disease.

### 4.4. Inflammatory Bowel Disease

Inflammatory bowel disease (IBD), which predominantly encompasses ulcerative colitis (UC) and Crohn’s disease, is a collective chronic intestinal inflammatory disease. Clinical studies have shown that the tryptophan metabolism might be associated with the severity of IBD [55]. A tryptophan deficiency might contribute to the development of IBD and/or aggravate a disease’s activity/severity [55]. In addition, a strong correlation between kynurenine levels and IBD has been demonstrated, which would provide new insights into IBD pathogenesis [56]. The modeling of the tryptophan metabolite fluxes in IBD has indicated that changes in gene expression shifted the intestinal tryptophan metabolism from the synthesis of serotonin towards the kynurenine pathway [57]. The enhancement of tryptophan processing via the kynurenine pathway in a pathological condition may also contribute to the progression of IBD. However, it has been suggested that the activation of the kynurenine pathway might be a part of the physiological mechanism, in order to compensate for the disease’s conditions [19,58].

### 4.5. Osteoporosis

Osteoporosis is also associated with the abnormalities of the kynurenine pathway [59]. Osteoporosis is a highly prevalent disease which is characterized by a low bone mass and represents the most common cause for bone fractures in the elderly [60]. A factor for the pathogenesis in osteoporosis may be the imbalanced activities of osteoblasts and osteoclasts [61]. Recent studies of T cells and osteoporosis have suggested the involvement of some T-cell subsets, regulatory T (Treg) cell and T helper 17 (Th17) cell subsets [62]. For example, the receptor activator of nuclear factor-κ B ligand (RANKL), a factor that is necessary for the differentiation of osteoclasts, may be released from Th17 cells [63]. The correlation between the osteoclasts and the ratio of the Th17/Treg cells in the immune system has attracted great attention in osteoporosis. In addition, the imbalance between Th17 cells and Treg cells may be a significant cause of atherosclerosis [64], which may be also linked to osteoporosis. The indoleamine 2,3-dioxygenase, which is the rate-limiting enzyme for the tryptophan catabolism through the kynurenine pathway, may be an important protein that controls the ratio of Th17/Treg cell balance [65]. A metabolite of indoleamine 2,3-dioxygenase could also inhibit the differentiation of Th17 cells and/or induce Treg cells, which may influence the Th17/Treg balance [66].

### 4.6. Polycystic Ovary Syndrome

Polycystic ovary syndrome (PCOS) is an intricate metabolic disorder that is regarded in women of reproductive age. Low-grade inflammatory situations, such as obesity and a compromised glucose tolerance, may act as collective metabolic disturbances in females with PCOS. Furthermore, the elevations of inflammatory cytokines, such as chemokines and/or interleukins, may worsen this metabolic disturbance in inflammatory disease patients with PCOS [67]. The pathology of PCOS may be a multifactorial disorder in the ovarian folliculogenesis, compromised gonadotropin levels, a genetic predisposition, and/or a gut microbiota imbalance. Inflammation appears to be the shared property between PCOS and the kynurenine pathway [68]. Extremely elevated metabolites have been thoroughly related to an increased risk of PCOS. Consequently, a combination of tryptophan, kynurenic acid, and/or kynurenine could be employed as a potential valuable marker to forecast the risk of PCOS. Additionally, the plasma level of the variations in the kynurenic acid and/or quinolinic acid may affect the risk of obesity in PCOS [69]. As a side note, feeding with excess L-tryptophan during the gestation of mothers may explain a reduction in fetal growth and/or the higher death rate of pups [70].

## 5. Kynurenine Metabolites Involved in Brain Memory System and Immunity

Tryptophan may be altered into several bioactive molecules, such as serotonin in the brain. However, more than 90% of tryptophan might be altered into kynurenine, in a process recognized as the kynurenine pathway [71]. Only a small amount of tryptophan may be metabolized into serotonin. In general, serotonin has been well-known as a monoamine neurotransmitter that could play an important role in several complex biological functions. In the brain, the biological roles of serotonin may include the functions of learning and/or memory [71,72]. In addition, it has been reported that kynurenine metabolites could also play roles in memory via the regulation of neurotransmitter systems. For instance, kynurenine and quinolinic acid (QA) have contrasting effects on NMDA receptor function [73]. Quinolinic acid may contribute to the stimulation of the NMDA receptor by inhibiting glutamate re-uptake [74]. Likewise, quinolinic acid has been revealed to activate the release of glutamate from neurons, preventing its re-uptake by astrocytes [75]. It has been demonstrated that quinolinic acid could result in substantial damage to learning in animal models [76]. Furthermore, stress could induce an increase in microglial numbers with a shift toward an inflammatory outline, which have an impaired crosstalk with neurons and impact the downregulation of glutamate signaling. Moreover, microglial immune responses after stresses could alter the kynurenine pathway through metabolites that might also modify glutamatergic transmission [77]. Stress is widely known to affect memory and synapses [78], which may be mediated by microglia. Microglia are involved in developmental neurogenesis and/or apoptosis in the hippocampus during the neonatal period [79]. Therefore, alterations in the microglia function may have long-lasting consequences for brain structure, neuronal function, and memory.

In the brain, the tryptophan metabolite kynurenic acid functions as an endogenous antagonist of glycine co-agonist sites in the NMDA receptor at endogenous brain concentrations [80]. The NMDA receptor is critical for the regulation of synaptic plasticity and/or several cognitive functions [81]. In particular, NMDA receptors in the hippocampal CA3 are required for the artificial association of memory events that are stored in the CA3 cell ensembles [82]. In addition, the NMDA-receptor-mediated structural plasticity of dendritic spines plays an important role in synaptic transmission in the brain during learning and memory formation [83], which is an important role for the endogenous kynurenic acid in the control of GABAergic neurotransmission in the prefrontal cortex [84]. Notably, increased levels of cerebral kynurenic acid have been detected in the aged brain and in several major neurological and psychiatric diseases [85]. Kynurenic acid has been also suggested to be related to cognitive impairments [86], which have been frequently discussed as possible markers of perioperative neurocognitive disorders [87]. A lower level of kynurenic acid has been associated with worse cognitive functioning in a bipolar depression group [88] (Figure 2).

Kynurenic acid is also a ligand of the GPR35 [89] and is able to activate the AHR [89]. The GPR35 is massively expressed in immune cells such as eosinophils, monocytes, natural killer-like T cells, and/or gastrointestinal cells, suggesting that the GPR35 might be physiologically important in these cells [90]. In addition, the GPR35 is upregulated in activated neutrophils, which could promote their migration [91]. It has been revealed that serotonin metabolite 5-hydroxyindoleacetic acid (5-HIAA) might be a ligand of the GPR35 [91]. Interestingly, tryptophan raised the levels of kynurenine, kynurenic acid, and 5-HIAA in a time-dependent manner [92]. The AHR is a transcription factor that incorporates dietary, microbial, metabolic, and/or environmental signals to control intricate cellular performances. The postbiotics of the gut microbiota metabolism may be a significant source of nutritional AHR ligands [93]. Short-chain fatty acids (SCFAs) are typical examples of postbiotics that may include acetate, butyrate, and propionate, which are derived from the fermentation of dietary fibers by gut microbiota that could activate the AHR [94]. The AHR pathway may be involved in several immune processes which are dynamic for the host’s intestinal homeostasis, as well as for the optimal microbiome [95]. In addition, the AHR might participate in innate immune responses to the microbial invasion of barrier tissues [96]. Furthermore, the AHR could modify the differentiation and function of both CD4^+^ T cells and CD8^+^ T cells [97], which might be involved in chronic autoimmune damage to CNS neurons [98] (Figure 2).

Some of the distinctive kynurenine aminotransferases could catalyze the transamination of L-kynurenine (L-KYN) to kynurenic acid [99]. Kynurenic acid can be formed from D-kynurenine (D-KYN) through an oxidative deamination by D-amino acid oxidase [100], which may account, in part, for the kynurenic acid synthesis from d-kynurenine in the brain [100]. Therefore, de novo kynurenic acid formation could involve different mechanisms. In particular, ROS should be considered as a possible alternative for the production of kynurenic acid from both L-KYN and D-KYN under physiological and/or pathological conditions [101]. Specifically, D-amino acid oxidase and ROS could function as alternative routes for the production of kynurenic acid [101]. Additionally, D-KYN could serve as the bioprecursor of kynurenic acid and 3-hydroxykynurenine, which may play a key role in the pathophysiology of several neurological and psychiatric diseases [102]. In addition, L-KYN also plays an imperative role as an immune check-point molecule, which might regulate several immune responses via the AHR [103]. L-KYN might be a potential marker of immune suppressant disorders, and is a metabolite of the kynurenine pathway [104].

Microglia have the ability to mediate innate immune memory and can be reprogrammed by primary stimuli to enhance or inhibit their immune response to secondary stimuli. Inflammatory stimulation is an important factor for microglia to mediate innate immune memory. Single or repeated stimulation can induce the microglia to form different phenotypes. This microglia-mediated innate immune response is involved in the regulation of immune memory. The neuronal deletion of excitatory amino acid transporter 2 may lead to the dysregulation of the kynurenine pathway, and the astrocytic deficiency of excitatory amino acid transporter 2 may also result in the dysfunction of innate and adaptive immune pathways, which correlate with cognitive decline [105]. Innate immune memory is closely associated with neurodegenerative diseases, brain tumors, brain damage, and psychosis. Further study on the mechanism of microglia-mediated innate immune memory is helpful to understand the occurrence and development of central nervous system diseases, and to provide new options for the treatment of central nervous system diseases.

## 6. Future Perspectives

The effects of various stressors on memory have been frequently reported [106]. Microglia might play a determinant role in traumatic memories, although their fundamental mechanisms still remain unidentified. As stress effects on both microglia and memory might be dependent on the duration of stress exposure time, it is imperative to know how microglia are involved in the properties of stress contact on the specific memory formation. The immune system and CNS might cooperate on several levels within a body. Here, it has been revealed that the brain keeps the evidence of certain inflammation that has happened in the body [107], which appears to be an immunological memory called “engrams” [108]. The associations of engrams may be reliant on environmental circumstances [109]. As a result, immunological engrams could bring back the inflammatory situation if it starts up again [108]. Created by repetitive inflammatory situations, these engrams might commit to a gentle progression of immune-related diseases [110] (Figure 3). In one possible way for therapeutic intervention, synaptic removal could be accomplished by microglia that are capable of introducing a lack of memories with engram cells [111]. It is deliberated that microglia could induce synapse elimination as a mechanism for disremembering memory holdings [111]. It has been described that microglia are interconnected to the value of synapse density, learning, and/or memory [112]. There are noteworthy associations between gut microbiota and demyelination by the microglia in the brain, suggesting that the crosstalk of gut microbiota and nerve function, based on the microglia, might play a key role for the clearance of engrams [113]. Modifications in the composition of gut microbiota with a number of pathological microorganisms might play a crucial role in the pathogenesis of several diseases. Consistently, growing evidence has also linked gut dysbiosis to the exacerbation of compromised autophagy in certain inflammations [114]. Remarkably, metformin, a pleiotropic drug that is capable of modulating autophagy, could modify the metabolic process of gut microbiota [115]. Accordingly, a dietary approach could adjust the gut microbiota that are potentially advantageous for the treatment of neurodegenerative disorders [116]. These gut microbiota might also inhibit the production of ROS, in order to maintain the health of the host brain [117]. It might be significant to reduce the levels of ROS for the improvement of neuro-regeneration with neuronal stem cells [118,119]. In addition, ROS might bias the task of the microglia with the oxidation of the mitochondria in glial cells [120]. Oxidative stress, inflammatory factors, and/or the alteration of microglia may altogether limit neuroplasticity in the CNS [121]. In addition, certain gut microbiota with SOD activity could probably prevent the incidence of neurodegenerative disorders by reducing the overproduction of ROS, which may clear the engram memory via the alteration of the functional microglia in the brain. The gut microbiome has been identified as a potential key factor for the kynurenine metabolism [122]. An antibiotic-induced microbial depletion could upsurge the utilization of the circulating tryptophan and decrease the kynurenine metabolism in the peripheral nerves [123]. Notably, the utilization of the tryptophan and kynurenine metabolism could recover the gut microbiome [124]. Probiotics may influence the ratio of kynurenine/tryptophan in the kynurenine metabolism [125]. In fact, *Lactobacillus plantarum* 299v can reduce the kynurenine concentration in the serum of patients with major depression and improve their cognitive function [126]. In addition, the consumption of *Bifidobacterium infantis* may also decrease the metabolism from tryptophan to kynurenine, increasing the concentration of the tryptophan in rat models with depression [127]. Similarly, *Lactobacillus johnsonii* could decrease the activity of indoleamine 2,3-dioxygenase in HT-29 intestinal epithelial cells in vitro [128]. Remarkably, we presume that cardiovascular diseases, osteoporosis, neurodegenerative diseases, and other immune-related diseases might similarly relate to the pathogenesis that is based on the engram memory system [116,129,130]. Future work should intensively focus on the anti-inflammatory effects of microbiota-derived tryptophan metabolites on host immune cells. Furthermore, it will be interesting to define how these pathways precisely interfere to regulate the host’s immune-related pathology on molecular levels.

## Figures and Tables

**Figure 1 ijms-24-05742-f001:**
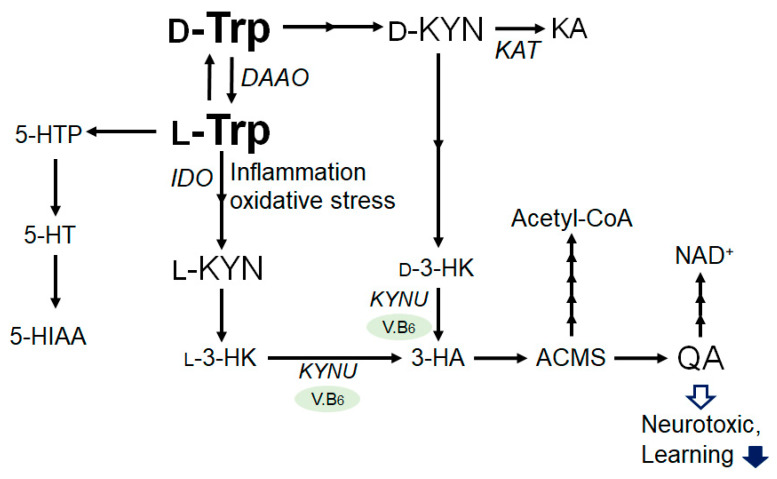
Illustration of the general L-Tryptophan and L-Kynurenine or D-Tryptophan and D-Kynurenine metabolic pathway in bacteria and/or in mammals. Some enzymatic degradation has also been shown. The arrowhead means stimulation and/or augmentation. As a footnote, some critical events have been omitted for simplicity. D-Trp: D-Tryptophan; L-Trp: L-Tryptophan; D-KYN: D-Kynurenine; L-KYN: L-Kynurenine; KA: Kynurenic acid; l-3-HK: l-3-Hydroxykynurenine, d-3-HK: d-3-Hydroxykynurenine; 3-HA: 3-Hydroxyanthranilic acid; ACMSD: α-amino-β-carboxymuconate-ε-semialdehyde; QA: Quinolinic acid; NAD: Nicotinamide adenine dinucleotide; 5-HTTP: 5-Hydroxytryptophan; 5-HTP: 5-Hydroxytryptophan; 5-HT: 5-hydroxytryptamine (Serotonin); 5-HIAA: 5-Hydroxyindoleacetic acid; V.B6: vitamin B6; DAAO: d-amino acid oxidase; IDO: Indoleamine 2,3-dioxygenase; KAT: kynurenine aminotransferase; and KYNU: kynureninase.

**Figure 2 ijms-24-05742-f002:**
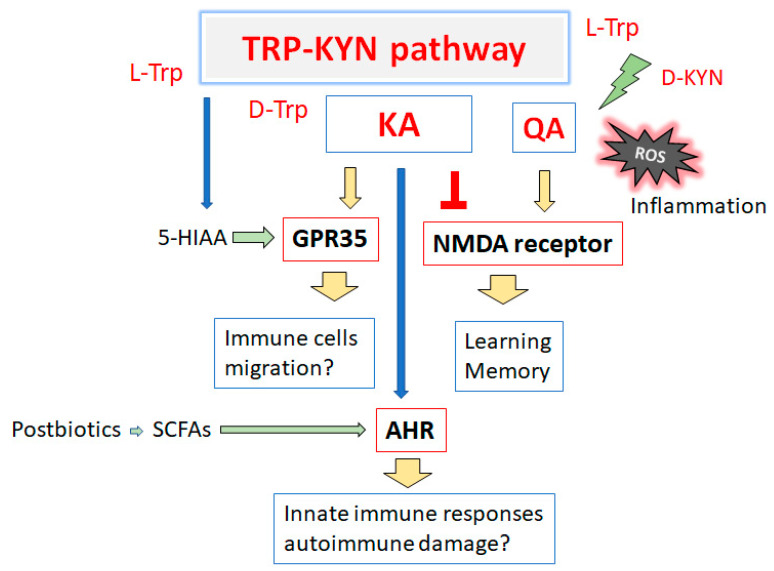
Schematic diagram for the biological effects of kynurenine metabolites from the tryptophan (TRP) and kynurenine (KYN) pathway. NMDA receptor, GPR35, and AHR might play key roles in the biological consequences. The arrowhead means stimulation and/or augmentation. Some critical events have been omitted for clarity. AHR: aryl hydrocarbon receptor; D-Trp: D-Tryptophan; L-Trp: L-Tryptophan; D-KYN: D-Kynurenine; L-KYN: L-Kynurenine; GPR35: G protein-coupled receptor 35; KA: Kynurenic acid; QA: Quinolinic acid; 5-HTP: 5-Hydroxytryptophan; 5-HT: 5-hydroxytryptamine (Serotonin); 5-HIAA: 5-Hydroxyindoleacetic acid; NMDA: N-methyl-D-aspartate; and SCFAs: short-chain fatty acids.

**Figure 3 ijms-24-05742-f003:**
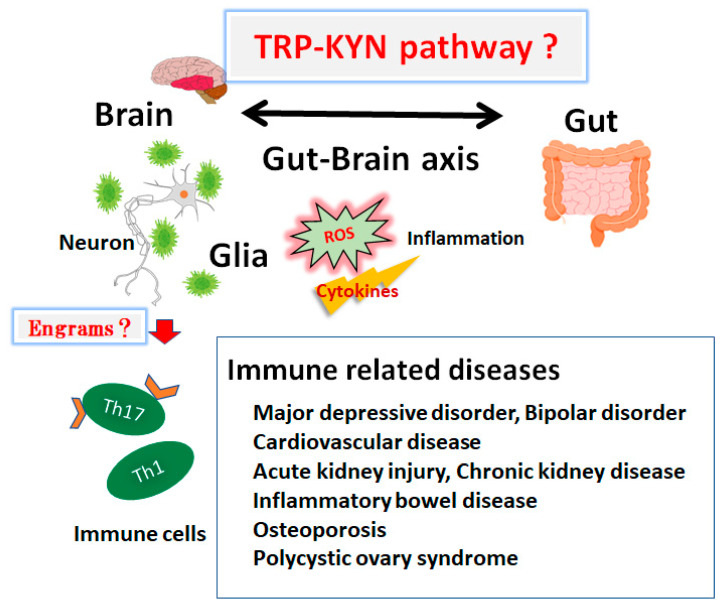
Schematic overview for the pathogenesis of immune-related diseases such as major depressive disorder, bipolar disorder, cardiovascular diseases, acute kidney injury, chronic kidney disease, inflammatory bowel disease, osteoporosis, and polycystic ovary syndrome. The gut–brain axis with tryptophan (TRP) and kynurenine (KYN) pathway might contribute to the pathogenesis of immune-related diseases via the formation of several “Engrams” in brain. Inflammation with several cytokines and/or reactive oxygen species (ROS) may be also involved in the modification of immune cells. Note that several important activities such as cytokine induction or anti-inflammatory reaction have been omitted for clarity.

## Data Availability

Not applicable.

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
