# Peer review of "The Tryptophan and Kynurenine Pathway Involved in the Development of Immune-Related Diseases"

_ijms, 2023, doi:10.3390/ijms24065742_

Round 1
Reviewer 1 Report
This work describes the related research of immune diseases involving tryptophan and tryptophan, analyzes the relationship with several conventional immune-related diseases and brain memory system, summarizes the related research of brain memory and puts forward the prospect. Although the aim of the study has certain application prospects, the research has sort lack of content with conclusion proving. And there are still some problems in the work of this article that need to be explained or corrected.
1. There are two ways to metabolize tryptophan, and the proportion of each way should be explained. Among them, what is the proportion of canine uric acid emphasized in the article.
2. This work has paid attention to the relationship between tryptophan and canine uric acid and some diseases. But it is mainly based on literature description and summary, results in the lacking quantitative analysis of relevant contents and the difficult to directly explain the importance of the two to different diseases. More statistical data can explain the problem better. For example, the variation of tryptophan and kynurenine pathway and the different populations (gender, age, race, basic disease, etc.) with different immune diseases or no. These data can be obtained from relevant institutions or articles mentioned in this work.
3. The "6. Future perspectives" section mainly focuses on the research, summary and outlook of the report on the memory system. The article gives a very complete overview of the memory system. However, the article lacks a summary of the aforementioned cardiovascular disease, Osteoporosis and other immune diseases. It is suggested to add relevant content or add a section of "conclusion" in the article.
Author Response
Reviewer1
This work describes the related research of immune diseases involving tryptophan and tryptophan, analyzes the relationship with several conventional immune-related diseases and brain memory system, summarizes the related research of brain memory and puts forward the prospect. Although the aim of the study has certain application prospects, the research has sort lack of content with conclusion proving. And there are still some problems in the work of this article that need to be explained or corrected.
Thank you so much for the good evaluation to our manuscript.
- There are two ways to metabolize tryptophan, and the proportion of each way should be explained. Among them, what is the proportion of canine uric acid emphasized in the article.
More than 90% of tryptophan is altered into kynurenine in a process known as kynurenine pathway, which has been mentioned at the beginning of section 5. As far as we have looked for anymore, the appropriate fine data for the proportion could not be found in the literature. Do you know anything about it?
- This work has paid attention to the relationship between tryptophan and canine uric acid and some diseases. But it is mainly based on literature description and summary, results in the lacking quantitative analysis of relevant contents and the difficult to directly explain the importance of the two to different diseases. More statistical data can explain the problem better. For example, the variation of tryptophan and kynurenine pathway and the different populations (gender, age, race, basic disease, etc.) with different immune diseases or no. These data can be obtained from relevant institutions or articles mentioned in this work.
True, that makes sense. However, it is too difficult for us at this stage to revise the manuscript with the statistical data and/or the result of quantitative analyses. Besides, there are few convincing evidences at present. We would like to describe them in the future opportunity.
- The "6. Future perspectives" section mainly focuses on the research, summary and outlook of the report on the memory system. The article gives a very complete overview of the memory system. However, the article lacks a summary of the aforementioned cardiovascular disease, osteoporosis and other immune diseases. It is suggested to add relevant content or add a section of "conclusion" in the article.
According to this suggestion, the "6. Future perspectives" section has been improved with an explanation for the memory system and addtional 2 references.
Reviewer 2 Report
In the present manuscript, the authors have reviewed the current knowledge and findings on tryptophan and its metabolites in several immune-related diseases. Overall, the manuscript is well structured, presented, and written and summarises all the current findings and aspects on how tryptophan and its major metabolites are involved in diseases. They also provide a clear overview of where studies are currently heading and where they should focus. Although the review is not overly long, it is supported by a large number of references. I have only one comment on Figure 1, which should be expanded to include a more detailed description of the possible metabolites and the enzymes involved in their production. It would also be beneficial for understanding to include diseases related to tryptophan and kynurenine in the Figure 1.
Author Response
Reviewer2
In the present manuscript, the authors have reviewed the current knowledge and findings on tryptophan and its metabolites in several immune-related diseases. Overall, the manuscript is well structured, presented, and written and summarises all the current findings and aspects on how tryptophan and its major metabolites are involved in diseases. They also provide a clear overview of where studies are currently heading and where they should focus. Although the review is not overly long, it is supported by a large number of references. I have only one comment on Figure 1, which should be expanded to include a more detailed description of the possible metabolites and the enzymes involved in their production. It would also be beneficial for understanding to include diseases related to tryptophan and kynurenine in the Figure 1.
Thank you so much for the good evaluation to our manuscript. According to this suggestion, Figure 1 has been improved.
Round 2
Reviewer 1 Report
The value in the original manuscript is 95% of "more than 95% of tryptophan is converted into kyrene in a process known as kyrene path." and no relevant literature is cited. As a review, this is not appropriate. On the other hand, in the new version, 95% has been changed to 90% without any explanation.
Author Response
Exactly, you are right. According to this suggestion, we have replaced the reference 71 with new one. And, we have cited the new ref 71 at the end of the indicated sentence in the text, as “more than 90% of tryptophan might be altered into kynurenine in a process recognized as kynurenine pathway [71]”. Firstly, we have used the sentence of "more than 95% of tryptophan is converted into kynurenine in a process known as kynurenine path," under the recognition of a literature by Richard et al (Int J Tryptophan Res. 2009;2:45-60). After the reading of your 1st round comment to our original manuscript, however, we had noticed it might be more appropriate to describe that 95% should be changed to 90%. Because we found several literatures indicating that “more than 90% of the Trp intake may be metabolized by the Trp-kynurenine-glutarate pathway”. The followings are some examples.
a. Shibata K, Fukuwatari T. Organ Correlation with Tryptophan Metabolism Obtained by Analyses of TDO-KO and QPRT-KO Mice. Int J Tryptophan Res. 2016;9:1-7.
b. Terakata M, Fukuwatari T, Sano M, et al. Truly niacin deficiency in quinolinic acid phosphoribosyltransferase (QPRT) knockout mice. J Nutr. 2012;142:2148–53
c. Keszthelyi D, Troost FJ, Masclee AA. Understanding the role of tryptophan and serotonin metabolism in gastrointestinal function. Neurogastroenterol Motil. 2009;21:1239–49.
d. Richard DM, Dawes MA, Mathias CW, Acheson A, Hill-Kapturczak N, Dougherty DM. L-Tryptophan: Basic Metabolic Functions, Behavioral Research and Therapeutic Indications. Int J Tryptophan Res. 2009;2:45-60.
Round 3
Reviewer 1 Report
The manuscript has been revised as comments, and reach the standard of ijms.